# Studying and Modeling of the Extraction Properties of the Natural Deep Eutectic Solvent and Sorbitol-Based Solvents in Regard to Biologically Active Substances from *Glycyrrhizae* Roots

**DOI:** 10.3390/molecules25071482

**Published:** 2020-03-25

**Authors:** Nikolay Boyko, Elena Zhilyakova, Anastasiya Malyutina, Oleg Novikov, Dmitriy Pisarev, Rimma Abramovich, Olga Potanina, Simon Lazar, Praskovia Mizina, Rita Sahaidak-Nikitiuk

**Affiliations:** 1Belgorod National Research University, BelSU, 308015 Belgorod, Russia; ezhilyakova@bsu.edu.ru (E.Z.); malyutina_a@bsu.edu.ru (A.M.); 2Peoples’ Friendship University of Russia, RUDN University, 117198 Moscow, Russia; ole9222@yandex.ru (O.N.); juniper05@mail.ru (D.P.); abramovich-ra@rudn.ru (R.A.); potanina-og@rudn.ru (O.P.); sjl512am@yandex.ru (S.L.); 3All-Russian Scientific Research Institute of Medicinal and Aromatic Plants, ARSRIMAP, 117216 Moscow, Russia; mizina-pg@yandex.ru; 4National University of Pharmacy, NUPh, 61002 Kharkiv, Ukraine; sagaidak_rita@ukr.net

**Keywords:** *Glycyrrhizae* roots, licuroside, glycyram, natural deep eutectic solvent, sorbitol, dielectric constant, mathematical model

## Abstract

The purpose of this work was the studying and modeling of the extraction properties of the sorbitol-based natural deep eutectic solvent (NADES) and sorbitol-based solvents in regard to biologically-active substances (BASs) from *Glycyrrhizae* roots using theoretical fundamentals based on the laws of statistical physics, thermodynamics, and physical chemistry previously developed by us. In our studies, we used *Glycyrrhizae* roots, simple maceration, plant raw material:solvent ratio 1:10 *w/v*, temperature 25 °C, extraction time 24 h; standards of licuroside and glycyram; RP HPLC, differential scanning calorimetry, integral dielectric, impedance and conductivity spectroscopy method of analysis; the following solvents: sorbitol-based NADES sorbitol:malic acid:water (1:1:3 in molar ratio), a modified solvent based on NADES sorbitol:malic acid:water:glycerin (1:1:1:1 in molar ratio) and sorbitol-based solvents sorbitol:ethanol:water at different ratios. It has been found that regression equations for sorbitol-based solvents in coordinates predicted by the theory have a high value of determination coefficient that equals to R^2^_e_ = 0.993 for glycyram and R^2^_e_ = 0.976 for licuroside. It has been found that the extraction properties of sorbitol-based NADES with a dielectric constant (ε) equal to 33 ± 2 units are equivalent to those of the sorbitol:ethanol:water solvent with ε = 34 units, and the extraction properties of modified solvent based on NADES with ε = 41 ± 2 units are inferior to those of the sorbitol-ethanol-water solvents with maximum value of BASs yield with the dielectric constant range 40 ÷ 50 units. The theoretical fundamentals suggested provide a possibility for an explanation of the mechanism, quantitative description of the extraction properties of the solvent, and target search of the optimal solvent by its dielectric constant.

## 1. Introduction

Presently, the issue of describing and modeling the extraction process of biologically-active substances (BASs) from the plant raw material remains open and is only at the initial stage of its solving. Moreover, due to discovery of new solvents, such as supercritical fluids, hydrotropic solvents, ionic liquids, eutectic solvents, and other types [1,2,3,4], there is a significant necessity for development of a theory that will provide a possibility for explanation and describing the processes in the extraction system in order to model and forecast the optimal conditions for BAS extraction from the plant raw material including the choice of the solvent type, its temperature, and volume.

However, theoretical developments (mathematical models) suggested by some authors do not allow achieving simplicity in calculations or can be used for a limited class of objects [5,6]. For this reason, we have made an attempt to develop some theoretical fundamentals of the extraction process related to describing the equilibrium state of the BASs in the extraction system and also to describing the influence of the solvent’s dielectric constant on the equilibrium concentration of the polar BASs in the solvent.

For this purpose, we used the postulates of the kinetic molecular theory of matter and the laws of statistical physics, thermodynamics, and physical chemistry [7]. The reasoning of our theoretical developments is based on the following postulates of the kinetic molecular theory of matter: (1) all substances consist of molecules/atoms; (2) all molecules/atoms are in constant and random motion; and (3) there are attraction and repulsion forces between the molecules/atoms.

The first postulate of the kinetic molecular theory provides enough reason so that the extraction system can be considered to be a macroscopic system with a huge number of different molecules (in a first approximation, as BASs, solvent, and plant raw material) and, respectively, we can describe it by the laws of statistical physics and thermodynamics. However, until now, the extraction system has not been analyzed from this viewpoint. The second postulate lays behind the kinetic side of the extraction process and describes the diffusion phenomena by the first and second Fick’s laws and Einstein’s equation for the diffusion coefficient. The third postulate is laid behind the energetical processes and is determined the extraction properties of solvent, static and kinetic side of the extraction process, dissolution/non-dissolution, adsorption/desorption, physical state of the macroscopic systems (gas, liquid, and solid state) and is described by different laws of thermodynamics, physical chemistry, surface chemistry, etc. Nevertheless, until now, the mechanism of energetical processes in the extraction system has not been developed.

Taking into account all of the aforesaid, we have suggested the following working hypotheses: (1) the process of equilibrium distribution of the BAS molecules between the phases of the extraction system follows Boltzmann distribution law with discrete levels of molecular energy (or Fermi–Dirac quantum statistics) and (2) the influence of the solvent’s dielectric constant on the equilibrium concentration of the polar BASs may be explained by the intermolecular interaction energy in which this parameter is present.

These working hypotheses and mathematical models developed on their basis have been successfully tested on different types of BASs (chlorogenic acid, rutin, euglobals, licurosid, glycyrrhizinic acid, eugenol, hyperforin, and silibinin), plant raw materials (*Calendulae officinalis* flowers, *Eucalyptus viminalis* leaves, *Glycyrrhizae* roots, clove buds, *Hypericum perforatum* herbs, and *Silybum marianum* fruits), and solvents (ethanol with different concentrations, methanol, 1-propanol, 2-propanol, acetone, ethyl acetate, and their water solutions) [8,9,10,11,12,13,14,15,16].

The results obtained encouraged us to expand our studies to other types of solvents, in particular, to studying and modeling of the extraction properties of sorbitol-based solvents and a natural deep eutectic solvent (NADES) based on sorbitol, as the value of the dielectric constant of some solvents on its basis is described in the scientific literature [17,18]. 

At present, NADESs are widely used in science for extraction of the following types of BASs: flavonoids [19,20], alkaloids [21], anthocyanins [22], flavors [23], saponins [24], etc. [25]. Moreover, sorbitol and some NADESs based on it can be used as excipients for development of such a drug form as syrups. For this reason, we have chosen *Glycyrrhizae* roots as plant raw material for our studies and dry and liquid extracts from *Glycyrrhizae* roots are used as an expectorant in different countries. For example, *Glycyrrhizae radices* syrup is used for this purpose in the Russian Federation, and Natures Aid Herbal Mucus Cough Syrup and Covonia Herbal Mucus Cough Syrup are used in the United Kingdom, etc. [26,27].

Some articles describe two main formulations of sorbitol based NADESs: sorbitol:malic acid:water (1:1:3 in mole ratio), sorbitol:malic acid (1:1 in mole ratio) [25,28,29]. It should be noted no information has been found in the scientific literature about the studies of the extraction properties of sorbitol-based solvents and sorbitol-based NADESs relative to the main BASs from *Glycyrrhizae* roots.

The purpose of this work was studying and modeling of the extraction properties of sorbitol-based solvents and a natural deep eutectic solvent in regard to biologically active substances from *Glycyrrhizae* roots using theoretical fundamentals based on the laws of statistical physics, thermodynamics, and physical chemistry previously developed by us.

## 2. Results and Discussion

The experimental values of the melting point (T_m_) for sorbitol, malic acid, and the glass transition temperature (T_g_) for sorbitol-based NADES evaluated by DSC method are presented in Table 1.

As can be seen from the data presented in Table 1, the sorbitol-based NADES (row 3) with sorbitol:malic acid:water (1:1:1 mole ratio) content has low glass transition temperature (−56 °C) that is much less than for sorbitol with the melting point equal to 97 °C and for malic acid with the melting point equal to 130 °C. There is confirmed information about the status of this mixture as a natural deep eutectic solvent [25,28,29]. It is should be mentioned that the weight loss for the sample after the three runs was equal to 4% and the glass transition temperature decreased approximately by 3.0 °C. That is these results provide a possibility for this type of NADES use at a room temperature for the extraction of BASs from the plant raw material. 

The results of the experimental study of the dependency between BASs concentration in the extracts and solvent’s dielectric constant based on a mixture of sorbitol:ethanol:water in coordinates predicted by the theory are presented in Figure 1 and Figure 2.

As can be seen from the data presented in Figure 1 and Figure 2, the experimental dependency of BASs concentration in extracts on the solvent’s dielectric constant based on a mixture of sorbitol:ethanol:water has good approximation by the regression model with determination coefficient for glycyram R^2^_e_ = 0.993 and for licuroside R^2^_e_ = 0.976. These values of the determination coefficient for experimental data are greater than the critical (theoretical) value, which for experimental points *n* = 5 and acceptance probability *P* = 99% equals R^2^_t_ = 0.919 [30]. Thus, in equation R^2^_e_ > R^2^_t_ (R^2^_e_ = 0.993 > 0.976 > R^2^_t_ = 0.919) is satisfied at acceptance probability *P* = 99%. Hence, the developed mathematical model that simulates the dependency of BAS equilibrium concentration in the solvent on its dielectric constant is compatible with the experimental data, and the suggested working hypotheses can be regarded to be reasonable. 

From the graphs in Figure 1 and Figure 2, we can see that the optimal value of the dielectric constant, for which maximum BASs concentration is observed, is within the range of ε = 40 ÷ 50 units (1/ε = 0.025 ÷ 0.020). These results agree with the results of our previous work on studying and modeling of glycyram and licuroside concentration dependency from the dielectric constant of ethanol-water solutions. The maximum concentrations of BASs were observed for the following values of dielectric constant: for glycyram ε = 48 ± 5 units and for licuroside ε = 40 ± 4 units [16]. 

Figure 1 and Figure 2 also present the data on the extraction properties of sorbitol-based NADES (see row 6, Table 2 and mark NADES in Figure 1 and Figure 2). It is interesting to note that the obtained values of glycyram concentration in coordinates predicted by the theory, are situated on the regression line and have little difference from the value for sorbitol-based solvent (see row 2, Table 2), which has good correlation with the values of their dielectric constants that are close to each other (*ε_sorbitol_*_*solvent*_ = 34 and *ε_eutectic_* = 33 ± 2 units). By contrast to glycyram, the experimental value of licuroside concentration in coordinates predicted by the theory, is not situated on the regression line and has relatively high difference from the value for the same sorbitol-based solvent. Nevertheless, this result has a good correlation with a supposition that the process of BAS extraction from the plant raw material is significantly influenced by the solvent’s dielectric constant.

In addition to these results, we decided to carry out an additional study with a modified solvent based on NADES that has a dielectric constant within the optimal range of values (see row 7, Table 2, *ε* = 41 ± 2 units). As seen from the data in Figure 1 and Figure 2, the extraction properties of the modified solvent based on NADES have relatively high level as it was predicted by the theory (see mark NADES-mod in Figure 1 and Figure 2), but these values are still less than those for a mixture of sorbitol:ethanol:water with a dielectric constant *ε* = 40 ÷ 50 units. This difference can be explained by the unpredicted processes that have place in extraction system and are not included into our model. 

We are aware that our theoretical fundamentals have some clear limitations such as a necessity to know the medium dielectric constant, and for now, this parameter is not determined for all types of solvents, mixtures, and also do not give the possibility for theoretical prediction of the optimum value of solvent’s dielectric constant to achieve the maximum yield for certain types of BASs from the plant raw material, etc. Nevertheless, considerable progress has been made in modeling and describing the experimental data by the theoretical approach suggested by us. 

It is important to note that the theoretical fundamentals suggested make it possible to carry out a targeted search for an optimal type of the solvent by its dielectric constant values.

## 3. Materials and Methods

### 3.1. Reagents and Solvents

In our studies, *Glycyrrhizae* roots with a range of particle size 0.1–0.5 mm, manufactured by Liktravy LLC, Zhitomir, Ukraine, batch No. 20917, expiry date 09.2022, were used as plant raw material.

As main components of NADES and sorbitol-based solvents, the following were used: dietary sorbitol, Sladkiymir LLC, Noviy Novgorod, Russia, batch No. 030317, expiry date 04 March, 2020; D,L-malic acid, high purity, AllHim Company, Kharkiv, Ukraine, batch No. 49859, expiry date 12/2022; ethanol 95% vol. for pharmaceutical and medical purposes, Medhimprom LLC, Zheleznodorozhniy, Russia, batch No. 100718, expiry date 07/2023; glycerin high grade, NevaReactiv Company, Saint-Petersburg, Russia, batch No. 053, expiry date 04/2021; formic acid 85% wt. high grade, NevaReactiv Company, Saint-Petersburg, Russia, batch No. 168, expiry date 02/2022; *aqua purificata*.

As standards, we used the following substances: official reference standards of the State Pharmacopoeia of Ukraine licuroside and monoammonium glycyrrhizate (glycyram) with content ≥95.0%.

### 3.2. Methods of Analysis

#### 3.2.1. Reverse Phase High Performance Liquid Chromatography Method (RP HPLC)

The quantitative analyses of licuroside and glycyram in the extracts were carried out by RP HPLC method with the use of Agilent Technologies chromatograph, series Agilent 1200 Infinity, the USA. The process of analysis was carried out under the following conditions: mobile phase A was 1% water solution of formic acid; mobile phase B was ethanol 95% vol.; in the regime of linear gradient; chromatographic column: Supelco Ascentis express C18, length 100 mm, inner diameter 4.6 mm, particle size 2.7 μm; mobile phase velocity 0.5 mL/min; temperature of chromatographic column 35 °C; sample volume 1 μL; before the analysis the sample was diluted in ethanol 70% vol. The conditions of RP HPLC analysis were the same as in the work of Zhilyakova et al. [31]. The analytical wavelengths were 370 and 248 nm.

The main parameters of analytical method validation and RP HPLC system suitability for the determination of licuroside and glycyram are presented in Table 2.

The density of solvents was determined by a gravimetrical method according to general monograph GPhA.1.2.1.0014.15 of the State Pharmacopoeia of the Russian Federation [32].

#### 3.2.2. A Method for Obtaining NADES and Sorbitol-Based Solvents

The mass ratio of sorbitol-based solvents is presented below in Table 3 (rows 1–5). The content of NADES is sorbitol:malic acid:water (1:1:3 in mole ratio), the mass ratio of components is presented below in Table 3 (row 6). The content of the modified solvent based on NADES is sorbitol:malic acid:water:glycerin (1:1:1:1 in mole ratio), the mass ratio of components is presented below in Table 3 (row 7). 

Values of the dielectric constant for a mixture of sorbitol:ethanol:water (sorbitol-based solvents) were used from article [18]. The dielectric constant values for sorbitol-based NADES and modified solvent based on NADES were determined using a dielectric, impedance and conductivity spectroscopy method.

The content and dielectric constants of sorbitol-based solvents are presented in Table 3.

The prescribed quantity of ingredients according to formulations indicated above were weighed using an analytical balance, put into a hermetic flask, mixed, and put on the ultrasound bath (Bandelin electronic, Germany, Type RK 102 CH, 35 kHz, 120/480 W) with temperature 60–70 °C for 10–15 min until clear solution was obtained.

#### 3.2.3. A method for Extract Preparation 

In our studies, we used a method of simple maceration at the plant raw material:solvent ratio 1:10 *w/v*; temperature 25 ± 1 °C; maceration time 24 h.

Accuracy weighted 1.00 g of plant raw material was put into the flask, 10.0 mL of solvent was added, after which it was placed into the thermostat for 24 h. Then the extract was filtered and analyzed by RP HPLC method for licuroside and glycyram content.

#### 3.2.4. A Method for Thermal Analysis

In our studies, we used a differential scanning calorimetry (DSC) analysis. The DSC analysis of sorbitol and malic acid was carried out by SDT Q600 simultaneous TGA/DSC/DTA analyzer, TA Instruments Inc., the USA, heating from 25 °C to 150 °C into argon atmosphere at a rate of 5 °C·min^−1^. The DSC analysis of sorbitol-based NADES (the row 6 in Table 2) was carried out by STA 449 F1 Jupiter TGA/DSC simultaneous, Netzsch, Germany, heating from −90 °C to 50 °C into nitrogen atmosphere at a rate of 5 °C·min^−1^ with three runs. A photo of physical state and results of DSC method for sorbitol, malic acid, and sorbitol-based NADES are presented in Appendix A.

#### 3.2.5. A Method for Dielectric Constant Analysis

In our studies, we used a dielectric, impedance and conductivity spectroscopy method by Novotherm-HT 1200 system, Alpha-A analyzer with test interface ZG4, manufactured by Novocontrol Technologies GmbH and Co. KG, Germany. The frequency range covered by the sample cell from was 1 to 14 MHz, at electrode potential 0.05 V and temperature 25 °C.

### 3.3. Theory

The first working hypothesis suggests that the process of BASs distribution between the phases in the extraction system should be described by Boltzmann law with discrete energy levels of molecules (or quantum Fermi-Dirac statistics). A mathematical model developed based on this working hypothesis is presented in Equation (1):(1)$$\frac{n}{n_{0}}=\frac{C\cdot V}{m_{0}}=\frac{1}{1+\exp\left[\frac{\Delta \mu }{RT}+a\right]}$$
where:*n*, *n_0_* are quantity of BAS molecules in the liquid phase and in the extraction system in general, moles;*C* is BAS concentration in the solvent, g/mL;*V* is solvent volume, mL;*m*_0_ is BAS mass in the extraction system, g;Δ*μ* is the change of BAS molecules’ chemical potential at the transition from plant raw material to the solvent, J/mole;*R* is gas constant, 8.314 J/(K·mol);*T* is absolute temperature, K;*a* is constant.

The second working hypothesis suggests that the influence of the solvent’s dielectric constant on the equilibrium concentration of polar BAS may be explained by the intermolecular energy with this parameter in it and Equation (1). The algorithm and intermediate computations of the final form of the mathematical model developed using this hypothesis is presented in Appendix A. The final form of the mathematical model developed based on this working hypothesis is presented below as Equation (2): (2)$$\ln\left(\frac{m_{0}}{C\cdot V}-1\right)=\frac{\Delta \mu }{R\cdot T}=\frac{b}{\varepsilon^{2}}+\frac{d}{\varepsilon }+f$$
where:*b*, *d*, *f* are empirical constants;*ε* is a dielectric constant of the solvent. 

Therefore, Equation (2) presented above should give the second order polynomial in coordinates $\ln\left(\frac{m_{0}}{C\cdot V}-1\right)=f\left(\frac{1}{\varepsilon }\right)$, and the closeness of agreement of theoretical provisions in the form of working hypotheses 1 and 2 and the experiment can be verified by the regression analysis and mathematical statistics. In addition to the above, the following condition has to be met: a determination coefficient of the regressive model (R^2^_e_) should be more than the table value (R^2^_t_) or R^2^_e_ > R^2^_t_ [30].

## 4. Conclusions

Extraction properties of sorbitol-based NADES and sorbitol-based solvents in regard to biologically active substances from *Glycyrrhizae* roots have been studied and modeled using our previously developed theoretical fundamentals based on statistical physics, thermodynamics, and physical chemistry. It has been found that regression equations of dependency of concentration BASs from the dielectric constant for sorbitol-based solvents in coordinates predicted by the theory have a high value of determination coefficient that equals to R^2^_e_ = 0.993 for glycyram and R^2^_e_ = 0.976 for licuroside. It has been found that the extraction properties of a sorbitol-based NADES and a modified NADES have a good correlation with their values of the dielectric constant. Suggested theoretical fundamentals provide a possibility for a theoretical explanation of the mechanism, quantitative description of the extraction properties of the solvent, and target search for an optimal solvent type by its dielectric constant.

## Figures and Tables

**Figure 1 molecules-25-01482-f001:**
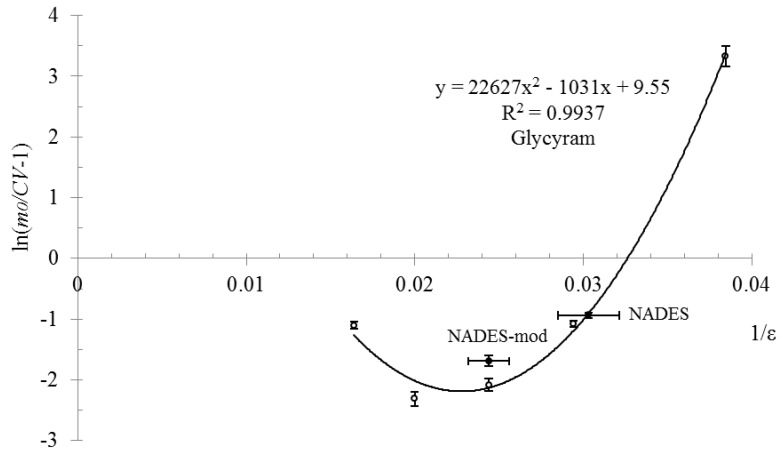
Dependency between glycyram concentration in the extracts and the dielectric constant of sorbitol-based solvents and NADESs in coordinates predicted by the theory.

**Figure 2 molecules-25-01482-f002:**
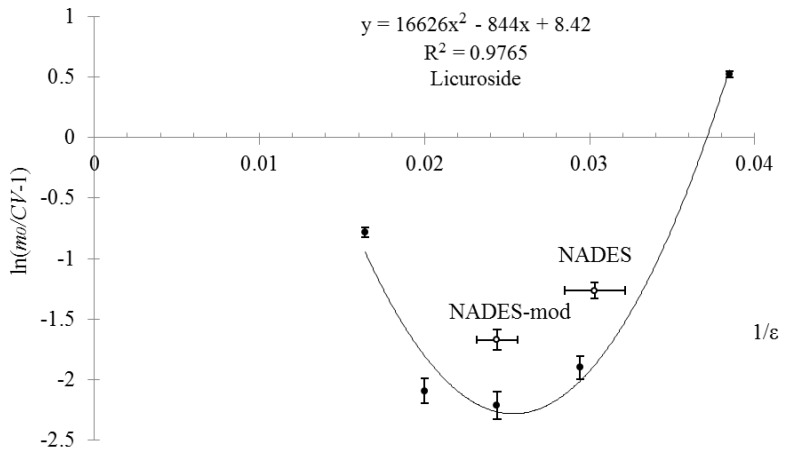
Dependency between licuroside concentration in the extracts and the dielectric constant of sorbitol-based solvents and NADESs in coordinates predicted by the theory.

**Table 1 molecules-25-01482-t001:** The experimental values of the melting point (T_m_) for sorbitol, malic acid, and the glass transition temperature (T_g_) for sorbitol-based NADES.

Compound	Mole Ratio	T_m_, °C	T_g_, °C
1. Sorbitol	-	97.3 ± 0.5	-
2. Malic acid	-	130.1 ± 0.5	-
3. Sorbitol-based NADES (sorbitol:malic acid:water)	1:1:3	-	−55.9 ± 1.5

* Note. The mean value and standard error of mean (X ± ΔX) were calculated at repeat counts *n* = 3 and significance level *p* = 0.95.

**Table 2 molecules-25-01482-t002:** Main parameters of analytical method validation and RP HPLC system suitability for licuroside and glycyram determination.

Parameter	Pharmacopoeia Limit [32]	Licuroside *	Glycyram *
1. Retention time, min	-	24.2 ± 0.2	37.3 ± 0.3 and 37.9 ± 0.3
2. Asymmetry parameter	0.8–1.5	0.82	0.84
3. Resolution between the peaks	≥1.5	2.5	1.7 and 1.5
4. Relative standard deviation, RSD, %	≤2.0	1.6	1.9 and 1.9
5. LOD, g/mL	-	2.0·10^−5^	8.9·10^−5^
6. LOQ, g/mL	-	6.1·10^−5^	2.7·10^−4^
7. Determination coefficient, *r*^2^	≥0.98	0.9999	0.9997
8. Linear regression equation, C(g/mL) = *f*(S(mAU·s))	-	*C* = (3.36 ± 0.04)·10^−7^·*S*	*C* = (1.77 ± 0.06)·10^−6^·*S*

* Note. The mean value and standard error of mean (X ± ΔX) were calculated at repeat counts *n* = 3 and significance level *P* = 0.95.

**Table 3 molecules-25-01482-t003:** The content and dielectric constants of sorbitol-based solvents.

No.	Content, % wt. *					Density, g/mL *	Dielectric Constant (*ε*)
	Sorbitol	Ethanol	Water	Malic Acid	Glycerin		
1	2.00 ± 0.01	91.0 ± 0.5	7.00 ± 0.04	0	0	0.815 ± 0.006	26
2	14.0 ± 0.1	66.0 ± 0.3	20.0 ± 0.1	0	0	0.900 ± 0.006	34
3	34.0 ± 0.2	42.0 ± 0.2	24.0 ± 0.1	0	0	1.018 ± 0.006	41
4	51.0 ± 0.3	22.0 ± 0.1	27.0 ± 0.1	0	0	1.130 ± 0.006	50
5	72.0 ± 0.4	0	28.0 ± 0.1	0	0	1.298 ± 0.006	61
6 **	49.0 ± 0.3	0	15.0 ± 0.1	36.0 ± 0.2	0	1.404 ± 0.006	33 ± 2 ***
7 **	42.7 ± 0.3	0	4.20 ± 0.03	31.5 ± 0.2	21.6 ± 0.1	1.381 ± 0.006	41 ± 2 ***

* Note. The mean value of the content and density (X ± ΔX) for the solutions obtained were calculated at repeat counts *n* = 3 and significance level *P* = 0.95. ** The content of sorbitol-based natural deep eutectic solvent (NADES) (row 6) is sorbitol:malic acid:water (1:1:3 in mole ratio), a modified solvent based on NADES (row 7) is sorbitol:malic acid:water:glycerin (1:1:1:1 in mole ratio). *** The dielectric constant values for sorbitol-based NADES (row 6) and modified solvent based on NADES (row 7) were determined experimentally (see Appendix A).

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
