# Peer review of "Studying and Modeling of the Extraction Properties of the Natural Deep Eutectic Solvent and Sorbitol-Based Solvents in Regard to Biologically Active Substances from Glycyrrhizae Roots"

_molecules, 2020, doi:10.3390/molecules25071482_

Round 1

Reviewer 1 Report

Most of the core work that should be included are briefly discussed and applied in the form of general models without explicitly describe how the authors take into account "the laws of statistical physics and thermodynamics" and "the energy processes of intermolecular interaction between the molecules of BAS, plant raw material skeleton, and the solvent...described by equations of Coulomb and Wan-der-Waals (in particular, Keesom and London." 

The authors give to the dielectric constant special attention, but the reasons based upon the abovementioned statements for this is absent. Moreover, the calculation of the dielectric constant of a DES is not trivial, as it seems to be the case here. See, for instance, "Fluid Phase Equilibria 448 (2017) 22-29" for details on the calculation of the DES´ static dielectric constant. 

Figure 1 summarizes most of the work; unfortunately, it is not easy to read. 

The authors should use a common type of biologically active substance for their experiments so that they could compare their results with other solvents besides NADEs. Additionally, regarding their choice of extract, most of the bibliography related to Glycyrrhizae radices syrup is not accessible. 

It is not clear, or at least there is no experimental evidence nor references, about the DES nature of the sorbitol-based solvents used in this investigation

Reviewer 2 Report

The presented work required some complement of NADES characterisation. First of all, the results related to transition temperatures investigated via eg. DSC technique are required. Sugar and carboxylic acids do not exhibit endothermic melting peak but glass transition - Tg and such systems are called Low Transition Temperature Mixtures - LTTM, so I suggest change ''NADES'' into ''LTTM''.

Francisco, A. van den Bruinhorst, M. C. Kroon, Green Chem. 2012, 14, 2153.

Viscosity values also should be presented, especially its influence is mentioned in line 220. This parameter also afect polysaccharide extraction.

line 87 NADES

line 114 change ''Density was determined..'' into ''Density of solvents was determined..''

Reviewer 3 Report

This manuscript describes the study and modeling of the extraction properties of natural deep eutectic solvent and sorbitol-based solvents in regard to biologically active substances from Glycyrrhizae roots using theoretical fundamentals based on laws of statistical physics, thermodynamics, physical, and colloid chemistry previously developed by the authors.
The results are original and the methods of analysis are well selected. I do not find important mistakes. Consequently, I recommend the manuscript for publication.

Author Response

Thanks a lot for your positive review!

Reviewer 4 Report

The manuscript by Boyko and co-workers entitled “Study and modeling of extraction properties of natural deep eutectic solvent and sorbitol-based 3 solvents in regard to biologically active substances from Glycyrrhizae roots” exploits the extraction properties of Deep Eutectic Solvents (DES) and sorbitol-based solvents. Which have technological and fundamental significance and is a useful contribution to the field.

The two points where this reviewer do believe the authors should re-examine their discussion is in the definition of DES and their mandatory validation (please check comments below) and the calculation of the dielectric constant using equation (1), mainly for the DES (“non-ideal” mixtures due to their “intrinsic” nature ruled by strong hydrogen bonding; please check comments below).

Also, in some parts the level of English does not meet the required standard. In short though, the manuscript may be appropriate for publication in the Molecules journal after a major revision, considering the previous comments and following points that should be addressed.

1) Page 1, Abstract. Check the abstract content. It should clearly summarize the content of the Manuscript (MS), experimental procedure and major outcomes/conclusions.

2) Page1, line 40. “Moreover, due to discovery of new extractants, such as supercritical, hydrotropic, ionic,…”, ionic? The authors are referring to ionic liquids.

3) Page 2, lines 64-73. The authors should detail the selection of NADES as extraction solvents. Based in previous work about DES, or at least ionic liquids, and not just base in the previous results of the authors with solvents and their aqueous solutions (results detailed in the previous paragraph).

4) Page 2, line 84. Check typo in particle size.

5) Page 2, line 87. Typo “NAES”. Check throughout the text.

6) Page 3, lines 117-120. This is a new DES? If so, the authors must validate that the mixture is a DES. If not, the authors must report the reference that indicate that the mixture is a DES. For the last case (DES validated in previous works), the authors must add this information and discussion in the introduction.

7) page 3, line 121. The authors calculated the dielectric constants of an an "ideal" mixture (equation (1)). The DES are "non-ideal" mixtures. The authors should determine the dielectric constant of the studied mixtures experimentally. At least this issue should be discussed in the MS. Additionally, in the cited reference [18] is stated otherwise, "Because of various complexities of the systems, however, no simple relationship appeared to exist between dielectric constants of the mixtures and those of the pure components.".

8) page 4, Table 2. The table must identify the mixtures that are NADES (6 and 7), and the ones that are "only" sorbitol-based solvents. This should be also detailed in the MS. Additionally, the molar ratio of NADES should be depicted in the table.

9) page 4, lines 134-137. The authors describe here the mixtures preparation method. But the DES must be validated. In case of DES reported in previous works, it must be clearly stated (check previous comments).

10) page 5, equation (3). check equation (3) deduction. ln(m0/(C.V) - 1)= Δμ/(RT) - a=.... Also, constant a in the equation (2) can't be the same that in equation (3).

11) Page 6, Figure 1. The different compounds must be better identified, in the figure caption or a legend in figure. The caption must express the experimental conditions.

12) Page 6, lines 205-206. Figure 1 must be more clearly represented. Data and its identification are confusing. NADES, NADES-mod and sorbitol-based solvents? The extracted (glycyram and Licuroside) must be better identified (e.g. dash& full lines).

13) Page 6, lines 215-216. Figure 1 is not well identified, and the attained conclusions can be misleading.

14) Page 6, lines 218-221. Check comment on the dielectric constant calculation. The DES are due to strong hydrogen bonding between the mixture components (“deep” depression on the mp). This is probably the main contribution to the different between the constant values.

Round 2

Reviewer 1 Report

The authors somewhat explained their rationale behind their hypothesis of the inclusion of dielectric constant to evaluate the suitability of a particular solvent for one specific type of natural solvate. While their brief explanation seems plausible, my opinion is that that part of the investigation, which is the foundation of the whole work, should be evaluated by an expert in the field of modeling and thermodynamics, not by a materials sciences reviewer.

From the viewpoint of materials science, the work is routinary and doest not merit publication in Molecules. 

Reviewer 2 Report

Please show DSC  for heat flow in Fig. S2. There are two tranitions changes one at near -13 and second at 17C. Why in Table 3 only first one is presented and discussed?  Please correct it before publication.   

I recommend to analyse DES by three-run DSC. heating-cooling-heating, especially, for the systems with water presence, please take into consideration it next time.

Reviewer 4 Report

In the revised manuscript the authors didn’t addressed properly the reviewers’ comments. Therefore, this reviewer rejects the publication in Molecules journal as it is.

Although, the manuscript may be appropriate for publication after a major revision, considering all the comments raised in the first revision.

The authors didn’t discuss properly the issues concerning the calculation of the dielectric constant and DES definition, and DES validation. For the last case, DES validation, the authors add DSC measurements. In table 3, the glass transition temperature of four compounds are reported, for three of them (compound 1,2 and 4), the reported values are the melting temperature. Additionally, compound nº 4 is classified as a deep eutectic solvent, and the depicted transition temperature states the opposite. Compound nº4 is not a DES.

Author Response

The answers for first revision were added.

The authors didn’t discuss properly the issues concerning the calculation of the dielectric constant and DES definition, and DES validation. For the last case, DES validation, the authors add DSC measurements. In table 3, the glass transition temperature of four compounds are reported, for three of them (compound 1,2 and 4), the reported values are the melting temperature. Additionally, compound nº 4 is classified as a deep eutectic solvent, and the depicted transition temperature states the opposite. Compound nº4 is not a DES.

Answer: The value of the dielectric constant was determined experimentally and the section results and discussion were changed according to the new information. Lines 248-264.

Dear reviewer we did not catch what does mean DES validation? As we understand it means that we should use the DSC method for found the glass transition temperature. We were repeated our studies at the bigger range of temperature and renew the data in table 3 and add new results into suppl.materials as Figure S6: Thermogram for a sorbitol-based NADES (sorbitol : malic acid : water with molar ratio 1:1:3). What exactly we should done else for DES definition or validation?

Round 3

Reviewer 4 Report

In the second revised manuscript the authors have determined the dielectric constants and modified accordingly the corresponding sections. Relatively to the DES definition, determination, validation and related experimental procedures, the authors still did not address properly the previous raised issues.

First, the authors must check the definition of DES. For example, “A DES is a fluid generally composed of two or three cheap and safe components that are capable of self-association, often through hydrogen bond interactions, to form a eutectic mixture with a melting point lower than that of each individual component” (Chem. Soc. Rev., 2012,41, 7108-7146).

Accordingly, in the previous revised version the compound nº 4 (Table 3) cannot be classified as a deep eutectic solvent (the depicted transition temperature states the opposite). Compound nº4 is not a DES. In the new revised version, the authors delete this compound from the table, but references to this compound as a NADES still appear in Table 2 and related discussion.

Also, the depicted transition temperatures for compounds nº 1 and nº 2 in Table 3 is not the glass transition temperature, it is the melting temperature (Tf). For a DES, that is liquid in all the temperature ranger scanned in the DSC experiments (do not present liquid-solid and solid-liquid transitions), the “validation” can be can attain by comparison of the Tf of the pure compounds with the DES. Table 3 must have both entries, Tf and Tg.

For the only sorbitol-based NADES validated (sorbitol: malic acid : water) 1:1:3, the authors must address carefully the considerably high amount of water. In the DSC experiments for the DES 1:1:3 how many successive runs -9ºc to 50ºc the authors completed? The weight of the pan was checked (previous and after experiment)? KF titration after DES run to check water amount? Additionally, due to high amount of water the authors should consider using for example NMR to study the water effect in the DES.

Therefore, this reviewer rejects the publication in Molecules journal as it is. Although, the manuscript may be appropriate for publication after revision, considering all the comments raised.

Author Response

Introduction

Dear reviewer, whatever the result is, I want to thank you for a lots of work you have done. It was difficult to communicate with a specialist by indirect way using an intermediate language but it was very useful for authors!

In the second revised manuscript the authors have determined the dielectric constants and modified accordingly the corresponding sections. Relatively to the DES definition, determination, validation and related experimental procedures, the authors still did not address properly the previous raised issues.

 First, the authors must check the definition of DES. For example, “A DES is a fluid generally composed of two or three cheap and safe components that are capable of self-association, often through hydrogen bond interactions, to form a eutectic mixture with a melting point lower than that of each individual component” (Chem. Soc. Rev., 2012,41, 7108-7146).

 Accordingly, in the previous revised version the compound nº 4 (Table 3) cannot be classified as a deep eutectic solvent (the depicted transition temperature states the opposite). Compound nº4 is not a DES. In the new revised version, the authors delete this compound from the table, but references to this compound as a NADES still appear in Table 2 and related discussion.

Answer

Thanks for your note about the presence of information related to compound n4 and presented under table 2. We have deleted this information and do not mention it in further text.

 Also, the depicted transition temperatures for compounds nº 1 and nº 2 in Table 3 is not the glass transition temperature, it is the melting temperature (Tf). For a DES, that is liquid in all the temperature ranger scanned in the DSC experiments (do not present liquid-solid and solid-liquid transitions), the “validation” can be can attain by comparison of the Tf of the pure compounds with the DES. Table 3 must have both entries, Tf and Tg.

Answer

The data about the melting point for sorbitol and malic acid were added in table 3 as a separate column. We have changed the term transition temperature for these substances for the melting point. We also added some information to the discussion: It is should be mentioned that the weight loss for the sample after the three runs was equal to 4 % and the glass transition temperature decreased approximately by 3.0 °C. We hope that now it is clear for understanding that mixture sorbitol:malic acid:water (1:1:3 in mole ratio) with glass transition temperature -59 C is NADES.

For the only sorbitol-based NADES validated (sorbitol: malic acid : water) 1:1:3, the authors must address carefully the considerably high amount of water. In the DSC experiments for the DES 1:1:3 how many successive runs -9ºc to 50ºc the authors completed? The weight of the pan was checked (previous and after experiment)? KF titration after DES run to check water amount? Additionally, due to high amount of water the authors should consider using for example NMR to study the water effect in the DES.

Answer

We have added the number of runs to the material and methods: three runs.

We have added the information mentioned above to the results and discussion: It is should be mentioned that the weight loss for the sample after the three runs was equal to 4 % and the glass transition temperature decreased approximately by 3.0 °C.

We did not need to use the KF titration method for determination of the weight loss of sample (due to evaporation of water) because the DSC method gives us this information that was equal to 4 %.

In this work, we did not study the state of solvent molecules, for example, formation of hydrogen bonds between them, by means of nuclear magnetic resonance (NMR), because this information is not used in our simplified mathematical model. However, when it becomes possible, we will expand of our studies in this direction.

Therefore, this reviewer rejects the publication in Molecules journal as it is. Although, the manuscript may be appropriate for publication after revision, considering all the comments raised.

Submission Date

26 December 2019

Date of this review

17 Mar 2020 20:55:07

P.S.:

Thanks,

With best regards,

The corresponding author.
